# Gradient $\ell_1$ Regularization for Quantization Robustness

**Milad Alizadeh**[*2,1], **Arash Behboodi**[1], **Mart van Baalen**[1], **Christos Louizos**[1], **Tijmen Blankevoort**[1], and **Max Welling**[1]

[1]Qualcomm AI Research[†]
Qualcomm Technologies Netherlands B.V.
{behboodi,mart,clouizos,tijmen,mwelling}@qti.qualcomm.com
[2]University of Oxford
milad.alizadeh@cs.ox.ac.uk

## ABSTRACT

We analyze the effect of quantizing weights and activations of neural networks on their loss and derive a simple regularization scheme that improves robustness against post-training quantization. By training quantization-ready networks, our approach enables storing a single set of weights that can be quantized on-demand to different bit-widths as energy and memory requirements of the application change. Unlike quantization-aware training using the straight-through estimator that only targets a specific bit-width and requires access to training data and pipeline, our regularization-based method paves the way for "on the fly" post-training quantization to various bit-widths. We show that by modeling quantization as a $\ell_\infty$-bounded perturbation, the first-order term in the loss expansion can be regularized using the $\ell_1$-norm of gradients. We experimentally validate the effectiveness of our regularization scheme on different architectures on CIFAR-10 and ImageNet datasets.

## 1 INTRODUCTION

Deep neural networks excel across a variety of tasks, but their size and computational requirements often hinder their real-world deployment. The problem is more challenging for mobile phones, embedded systems, and IoT devices, where there are stringent requirements in terms of memory, compute, latency, and energy consumption. Quantization of parameters and activations is often used to reduce the energy and computational requirements of neural networks. Quantized neural networks allow for more speed and energy efficiency compared to floating-point models by using fixed-point arithmetic.

However, naive quantization of pre-trained models often results in severe accuracy degradation, especially when targeting bit-widths below eight (Krishnamoorthi, 2018). Performant quantized models can be obtained via quantization-aware training or fine-tuning, i.e., learning full-precision *shadow* weights for each weight matrix with backpropagation using the straight-through estimator (STE) (Bengio et al., 2013), or using other approximations (Louizos et al., 2018). Alternatively, there have been successful attempts to recover the lost model accuracy without requiring a training pipeline (Banner et al., 2018; Meller et al., 2019; Choukroun et al., 2019; Zhao et al., 2019) or representative data (Nagel et al., 2019).

But these methods are not without drawbacks. The shadow weights learned through quantization-aware fine-tuning often do not show robustness when quantized to bit-widths other than the one they were trained for (see Table 1). In practice, the training procedure has to be repeated for each quantization target. Furthermore, post-training recovery methods require intimate knowledge of the relevant architectures. While this may not be an issue for the developers training the model in the first

---

[*]Work done during internship at Qualcomm AI Research
[†]Qualcom AI Research is an initiative of Qualcomm Technologies, Inc.

place, it is a difficult step for middle parties that are interested in picking up models and deploying them to users down the line, e.g., as part of a mobile app. In such cases, one might be interested in automatically constraining the computational complexity of the network such that it conforms to specific battery consumption requirements, e.g. employ a 4-bit variant of the model when the battery is less than 20% but the full precision one when the battery is over 80%. Therefore, a model that can be quantized to a specific bit-width "on the fly" without worrying about quantization aware fine-tuning is highly desirable.

In this paper, we explore a novel route, substantially different from the methods described above. We start by investigating the theoretical properties of noise introduced by quantization and analyze it as a $\ell_\infty$-bounded perturbation. Using this analysis, we derive a straightforward regularization scheme to control the maximum first-order induced loss and learn networks that are inherently more robust against post-training quantization. We show that applying this regularization at the final stages of training, or as a fine-tuning step after training, improves post-training quantization across different bit-widths at the same time for commonly used neural network architectures.

## 2 FIRST-ORDER QUANTIZATION-ROBUST MODELS

In this section, we propose a regularization technique for robustness to quantization noise. We first propose an appropriate model for quantization noise. Then, we show how we can effectively control the first-order, i.e., the linear part of the output perturbation caused by quantization. When the linear approximation is adequate, our approach guarantees the robustness towards various quantization bit-widths simultaneously.

We use the following notation throughout the paper. The $\ell_p$-norm of a vector $\boldsymbol{x}$ in $\mathbb{R}^n$ is denoted by $\|\boldsymbol{x}\|_p$ and defined as $\|\boldsymbol{x}\|_p := (\sum_{i=1}^n |x_i|^p)^{1/p}$ for $p \in [1, \infty)$. At its limit we obtain the $\ell_\infty$-norm defined by $\|\boldsymbol{x}\|_\infty := \max_i |x_i|$. The inner product of two vectors $\boldsymbol{x}$ and $\boldsymbol{y}$ is denoted by $\langle \boldsymbol{x}, \boldsymbol{y} \rangle$.

### 2.1 ROBUSTNESS ANALYSIS UNDER $\ell_p$-BOUNDED ADDITIVE NOISE

The error introduced by rounding in the quantization operation can be modeled as a generic additive perturbation. Regardless of which bit-width is used, the quantization perturbation that is added to each value has bounded support, which is determined by the width of the quantization bins. In other words, the quantization noise vector of weights and activations in neural networks has entries that are bounded. Denote the quantization noise vector by $\boldsymbol{\Delta}$. If $\delta$ is the width of the quantization bin, the vector $\boldsymbol{\Delta}$ satisfies $\|\boldsymbol{\Delta}\|_\infty \leq \delta/2$. Therefore we model the quantization noise as a perturbation bounded in the $\ell_\infty$-norm. A model robust to $\ell_\infty$-type perturbations would also be robust to quantization noise.

To characterize the effect of perturbations on the output of a function, we look at its tractable approximations. To start, consider the first-order Taylor-expansion of a real valued-function $f(\boldsymbol{w} + \boldsymbol{\Delta})$ around $\boldsymbol{w}$:

$$f(\boldsymbol{w} + \boldsymbol{\Delta}) = f(\boldsymbol{w}) + \langle \boldsymbol{\Delta}, \nabla f(\boldsymbol{w}) \rangle + R_2, \tag{1}$$

where $R_2$ refers to the higher-order residual error of the expansion. We set $R_2$ aside for the moment and consider the output perturbation appearing in the first-order term $\langle \boldsymbol{\Delta}, \nabla f(\boldsymbol{w}) \rangle$. The maximum of the first-order term among all $\ell_\infty$-bounded perturbations $\boldsymbol{\Delta}$ is given by:

$$\max_{\|\boldsymbol{\Delta}\|_\infty \leq \delta} \langle \boldsymbol{\Delta}, \nabla f(\boldsymbol{w}) \rangle = \delta \|\nabla f(\boldsymbol{w})\|_1. \tag{2}$$

To prove this, consider the inner product of $\boldsymbol{\Delta}$ and an arbitrary vector $\boldsymbol{x}$ given by $\sum_{i=1}^n n_i x_i$. Since $|n_i|$ is assumed to be bounded by $\delta$, each $n_i x_i$ is bounded by $\delta|x_i|$, which yields the result. The maximum in Equation 2 is obtained indeed by choosing $\boldsymbol{\Delta} = \delta \operatorname{sign}(\nabla f(\boldsymbol{w}))$.

Equation 2 comes with a clear hint. We can guarantee that the first-order perturbation term is small if the $\ell_1$-norm of the gradient is small. In this way, the first-order perturbation can be controlled efficiently for various values of $\delta$, i.e. for various quantization bit-widths. In other words, an effective way for controlling the quantization robustness, up to first-order perturbations, is to control the $\ell_1$-norm of the gradient. As we will shortly argue, this approach yields models with the best robustness.

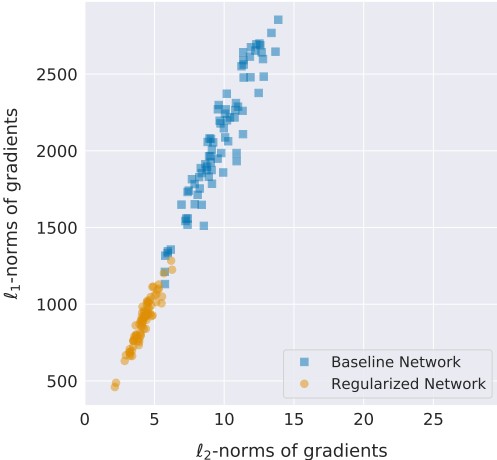 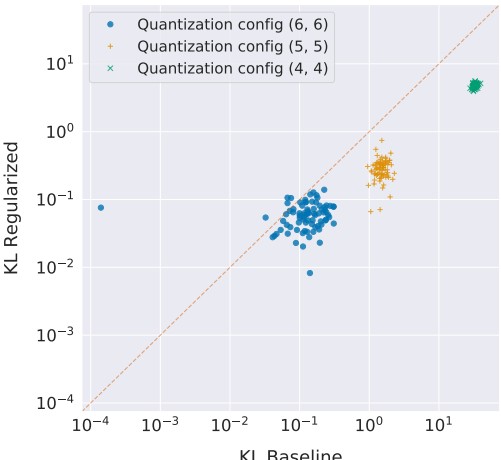

Figure 1: $\ell_1$- and $\ell_2$-norms of the gradients for CIFAR-10 test-set mini-batches. Note the difference between the scales on the horizontal and vertical axis. We observe that our regularization term decreases the $\ell_1$-norm significantly, compared to its unregularized counterpart.

Figure 2: KL-divergence of the floating point predictive distribution to the predictive distribution of the quantized model for CIFAR-10 test-set mini-batches. We observe that the regularization leads to a smaller gap, especially for smaller bit-widths.

This conclusion is based on worst-case analysis since it minimizes the upper bound of the first-order term, which is realized by the worst-case perturbation. Its advantage, however, lies in simultaneous control of the output perturbation for all $\delta$s and all input perturbations. In the context of quantization, this implies that the first-order robustness obtained in this way would hold regardless of the adopted quantization bit-width or quantization scheme.

The robustness obtained in this way would persist even if the perturbation is bounded in other $\ell_p$-norms. This is because the set of $\ell_\infty$-bounded perturbations includes all other bounded perturbations, as for all $p \in [1, \infty)$, $\|x\|_p \leq \delta$ implies $\|x\|_\infty \leq \delta$ (see Figure 8) . The robustness to $\ell_\infty$-norm perturbations is, therefore, the most stringent one among other $\ell_p$-norms, because a model should be robust to a broader set of perturbations. Controlling the $\ell_1$-norm of the gradient guarantees robustness to $\ell_\infty$-perturbations and thereby to all other $\ell_p$-bounded perturbations.

In what follows, we propose regularizing the $\ell_1$-norm of the gradient to promote robustness to bounded norm perturbations and in particular bounded $\ell_\infty$-norm perturbations. These perturbations arise from quantization of weights and activations of neural networks.

## 2.2 ROBUSTNESS THROUGH REGULARIZATION OF THE $\ell_1$-NORM OF THE GRADIENT

We focused on weight quantization in our discussions so far, but we can equally apply the same arguments for activation quantization. Although the activations are not directly learnable, their quantization acts as an additive $\ell_\infty$-bounded perturbation on their outputs. The gradient of these outputs is available. It therefore suffices to accumulate all gradients along the way to form a large vector for regularization.

Suppose that the loss function for a deep neural network is given by $L_{CE}(\mathbb{W}, \mathbb{Y}; \boldsymbol{x})$ where $\mathbb{W}$ denotes the set of all weights, $\mathbb{Y}$ denotes the set of outputs of each activation and $\boldsymbol{x}$ the input. We control the $\ell_1$-norm of the gradient by adding the regularization term

$$\sum_{\boldsymbol{W}_l \in \mathbb{W}} \|\nabla_{\boldsymbol{W}_l} L_{CE}(\mathbb{W}, \mathbb{Y}; \boldsymbol{x})\|_1 + \sum_{\boldsymbol{y}_l \in \mathbb{Y}} \|\nabla_{\boldsymbol{y}_l} L_{CE}(\mathbb{W}, \mathbb{Y}; \boldsymbol{x})\|_1$$

to the loss, yielding an optimization target

$$L(\mathbb{W}; \boldsymbol{x}) = L_{CE}(\mathbb{W}, \mathbb{Y}; \boldsymbol{x}) + \lambda_w \sum_{\boldsymbol{W}_l, \in \mathbb{W}} \|\nabla_{\boldsymbol{W}_l} L_{CE}(\mathbb{W}, \mathbb{Y}; \boldsymbol{x})\|_1 + \lambda_y \sum_{\boldsymbol{y}_l \in \mathbb{Y}} \|\nabla_{\boldsymbol{y}_l} L_{CE}(\mathbb{W}, \mathbb{Y}; \boldsymbol{x})\|_1,$$

(3)

where $\lambda_w$ and $\lambda_y$ are weighing hyper-parameters.

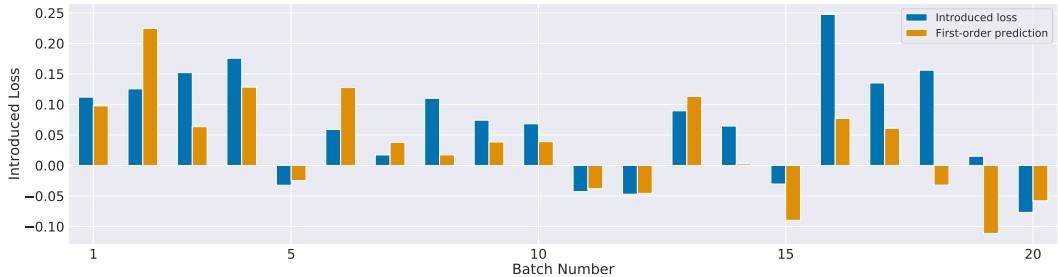

Figure 3: **Predicting induced loss using first-order terms.** We added $\ell_\infty$-bounded noise with $\delta$ corresponding to 4-bit quantization to all weights of ResNet-18 and compared the induced loss on the CIFAR-10 test-set with the predictions using gradients. While not perfect, the first-order term is not insignificant.

### 2.3 ALTERNATIVES TO THE $\ell_1$-REGULARIZATION

The equivalence of norms in finite-dimensional normed spaces implies that all norms are within a constant factor of one another. Therefore, one might suggest regularizing any norm to control other norms. Indeed some works attempted to promote robustness to quantization noise by controlling the $\ell_2$-norm of the gradient (Hoffman et al., 2019). However, an argument related to the curse of dimensionality can show why this approach will not work. The equivalence of norms for $\ell_1$ and $\ell_2$ in $n$-dimensional space is stated by the inequality:

$$\|\boldsymbol{x}\|_2 \leq \|\boldsymbol{x}\|_1 \leq \sqrt{n}\,\|\boldsymbol{x}\|_2.$$

Although the $\ell_2$-norm bounds the $\ell_1$-norm from above, it is vacuous if it does not scale with $1/\sqrt{n}$. Imposing such a scaling is demanding when $n$, which is the number of trainable parameters, is large. Figure 1 shows that there is a large discrepancy between these norms in a conventionally trained network, and therefore small $\ell_2$-norm does not adequately control the $\ell_1$-norm. A very similar argument can be provided from a theoretical perspective (see the supplementary materials).

To guarantee robustness, the $\ell_2$-norm of the gradient, therefore, should be pushed as small as $\Theta(1/\sqrt{n})$. We experimentally show in Section 4 that this is a difficult task. We therefore directly control the $\ell_1$-norm in this paper. Note that small $\ell_1$-norm is guaranteed to control the first order-perturbation for all types of quantization noise with bounded support. This includes symmetric and asymmetric quantization schemes.

Another concern is related to the consistency of the first-order analysis. We neglected the residual term $R_2$ in the expansion. Figure 3 compares the induced loss after perturbation with its first-order approximation. The approximation shows a strong correlation with the induced loss. We will see in the experiments that the quantization robustness can be boosted by merely controlling the first-order term. Nonetheless, a higher-order perturbation analysis can probably provide better approximations. Consider the second-order perturbation analysis:

$$f(\boldsymbol{w} + \boldsymbol{\Delta}) = f(\boldsymbol{w}) + \langle \boldsymbol{\Delta}, \nabla f(\boldsymbol{w}) \rangle + \frac{1}{2} \boldsymbol{\Delta}^T \nabla^2 f(\boldsymbol{w}) \boldsymbol{\Delta} + R_3.$$

Computing the worst-case second-order term for $\ell_\infty$-bounded perturbations is hard. Even for convex functions where $\nabla^2 f(\boldsymbol{w})$ is positive semi-definite, the problem of computing worst-case second-order perturbation is related to the mixed matrix-norm computation, which is known to be NP-hard. There is no polynomial-time algorithm that approximates this norm to some fixed relative precision (Hendrickx & Olshevsky, 2010). For more discussions, see the supplementary materials. It is unclear how this norm should be controlled via regularization.

## 3 RELATED WORK

A closely related line of work to ours is the analysis of the robustness of the predictions made by neural networks subject to an adversarial perturbation in their input. Quantization can be seen as a similar scenario where non-adversarial perturbations are applied to weights and activations instead. Cisse et al. (2017) proposed a method for reducing the network's sensitivity to small perturbations

by carefully controlling its global Lipschitz. The Lipschitz constant of a linear layer is equal to the spectral norm of its weight matrix, i.e., its largest singular value. The authors proposed regularizing weight matrices in each layer to be close to orthogonal: $\sum_{\mathbf{W}_l \in \mathbb{W}} \left\| \mathbf{W}_l^T \mathbf{W}_l - \mathbf{I} \right\|^2$. All singular values of orthogonal matrices are one; therefore, the operator does not amplify perturbation (and input) in any direction. Lin et al. (2019) studied the effect of this regularization in the context of quantized networks. The authors demonstrate the extra vulnerability of quantized models to adversarial attacks and show how this regularization, dubbed "Defensive Quantization", improves the robustness of quantized networks. While the focus of Lin et al. (2019) is on improving the adversarial robustness, the authors report limited results showing accuracy improvements of post-training quantization.

The idea of regularizing the norm of the gradients has been proposed before (Gulrajani et al., 2017) in the context of GANs, as another way to enforce Lipschitz continuity. A differentiable function is 1-Lipschitz if and only if it has gradients with $\ell_2$-norm of at most 1 everywhere, hence the authors penalize the $\ell_2$-norm of the gradient of the critic with respect to its input. This approach has a major advantage over the methods mentioned above. Using weight regularization is only well-defined for 2D weight matrices such as in fully-connected layers. The penalty term is often approximated for convolutional layers by reshaping the weight kernels into 2D matrices. Sedghi et al. (2018) showed that the singular values found in this weight could be very different from the actual operator norm of the convolution. Some operators, such as nonlinearities, are also ignored. Regularizing Lipschitz constant through gradients does not suffer from these shortcomings, and the operator-norm is regularized directly. Guo et al. (2018) demonstrated that there exists an intrinsic relationship between sparsity in DNNs and their robustness against $\ell_\infty$ and $\ell_2$ attacks. For a binary linear classifier, the authors showed that they could control the $\ell_\infty$ robustness, and its relationship with sparsity, by regularizing the $\ell_1$ norm of the weight tensors. In the case of a linear classifier, this objective is, in fact, equivalent to our proposed regularization penalty.

Finally, another line of work related to ours revolves around quantization-aware training. This can, in general, be realized in two ways: 1) regularization and 2) mimicking the quantization procedure during the forward pass of the model. In the first case, we have methods (Yin et al., 2018; Achterhold et al., 2018) where there are auxiliary terms introduced in the objective function such that the optimized weights are encouraged to be near, under some metric, to the quantization grid points, thus alleviating quantization noise. In the second case, we have methods that rely on either the STE (Courbariaux et al., 2015; Rastegari et al., 2016; Jacob et al., 2018), stochastic rounding (Gupta et al., 2015; Gysel, 2016), or surrogate objectives and gradients (Louizos et al., 2018; Shayer et al., 2017). While all of the methods above have been effective, they still suffer from a major limitation; they target one-specific bit-width. In this way, they are not appropriate for use-cases where we want to be able to choose the bit-width "on the fly".

## 4 EXPERIMENTS

In this section we experimentally validate the effectiveness of our regularization method on improving post-training quantization. We use the well-known classification tasks of CIFAR-10 with ResNet-18 (He et al., 2016) and VGG-like (Simonyan & Zisserman, 2014) and of ImageNet with ResNet-18. We compare our results for various bit-widths against (1) unregularized baseline networks (2) Lipschitz regularization methods (Lin et al., 2019; Gulrajani et al., 2017) and (3) quantization-aware fine-tuned models. Note that Gulrajani et al. (2017) control the Lipschitz constant under an $\ell_2$ metric by explicitly regularizing the $\ell_2$-norm of the gradient, while Lin et al. (2019) essentially control an upper bound on the $\ell_2$-norm of the gradient. Comparing against these baselines thus gives insight into how our method of regularizing the $\ell_1$-norm of the gradient compares against regularization of the $\ell_2$-norm of the gradient.

### 4.1 EXPERIMENTAL SETUP

**Implementation and complexity** Adding the regularization penalty from Equation 3 to the training objective requires higher-order gradients. This feature is available in the latest versions of frameworks such as Tensorflow and PyTorch (of which we have used the latter for all our experiments). Computing $\nabla_{\mathbf{w}} \|\nabla_{\mathbf{w}} \mathcal{L}\|_1$ using automatic differentiation requires $O(2 \times C \times E)$ extra computations, where $E$ is the number of elementary operations in the original forward computation graph, and $C$

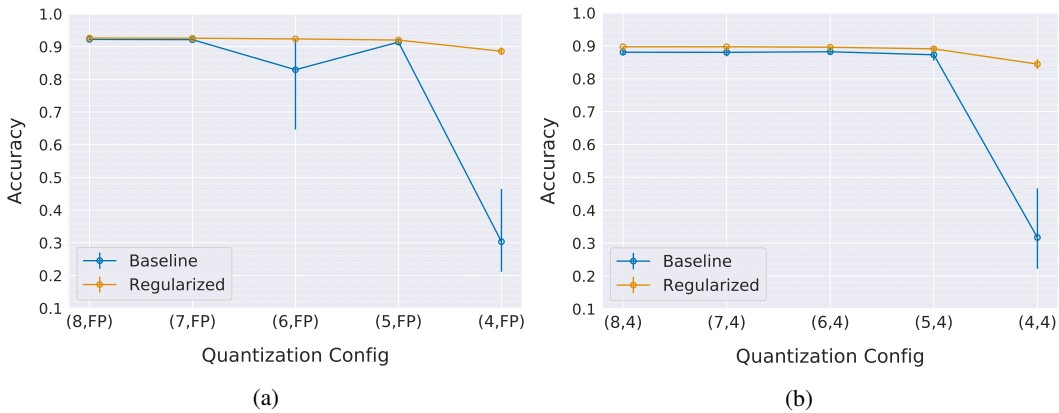

Figure 4: **Accuracy of regularized VGG-like after post-training quantization.** We trained 5 models with different initializations and show the mean accuracy for each quantization configuration. The error bars indicate min/max observed accuracies. (a) Weight-only quantization (b) Activation quantization fixed to 4-bits

is a fixed constant (Baydin et al., 2018). This can be seen from the fact that $\|\nabla_{\mathbf{w}}\mathcal{L}\|_1$ is a function $\mathbb{R}^{|\mathbf{w}|} \to \mathbb{R}$, where $|\mathbf{w}|$ denotes the number of weights and the computation of the gradient w.r.t. the loss contains $E$ elementary operations, as many as the forward pass. In practice, enabling regularization increased time-per-epoch time on CIFAR10 from 14 seconds to 1:19 minutes for VGG, and from 24 seconds to 3:29 minutes for ResNet-18. On ImageNet epoch-time increased from 33:20 minutes to 4:45 hours for ResNet-18. The training was performed on a single NVIDIA RTX 2080 Ti GPU.

However, in our experiments we observed that it is not necessary to enable regularization from the beginning, as the $\ell_1$-norm of the gradients decreases naturally up to a certain point as the training progresses (See Appendix D for more details). We therefore only enable regularization in the last 15 epochs of training or as an additional fine-tuning phase. We experimented with tuning $\lambda_w$ and $\lambda_y$ in Equation 3 separately but found no benefit. We therefore set $\lambda_w = \lambda_y = \lambda$ for the remainder of this section.

We use a grid-search to find the best setting for $\lambda$. Our search criteria is ensuring that the performance of the *unquantized* model is not degraded. In order to choose a sensible range of values we first track the regularization and cross-entropy loss terms and then choose a range of $\lambda$ that ensures their ratios are in the same order of magnitude. We do not perform any quantization for validation purposes during the training.

**Quantization details**   We use uniform symmetric quantization (Jacob et al., 2018; Krishnamoorthi, 2018) in all our experiments unless explicitly specified otherwise. For the CIFAR 10 experiments we fix the activation bit-widths to 4 bits and then vary the weight bits from 8 to 4. For the Imagenet experiments we use the same bit-width for both weights and activations. For the quantization-aware fine-tuning experiments we employ the STE on a fixed (symmetric) quantization grid. In all these experiments we perform a hyperparameter search over learning rates for each of the quantization bit-widths and use a fixed weight decay of $1e - 4$. For our experiments with defensive quantization (Lin et al., 2019) we perform a hyperparameter search over the scaling parameters of the regularizer and the learning rate. We limit the search over the scaling parameters to those mentioned in (Lin et al., 2019) and do not use weight decay. When applying post-training quantization we set the activation ranges using the batch normalization parameters as described in (Nagel et al., 2019).

When a model is fine-tuned to a target bit-width and evaluated on a higher bit-width, we can trivially represent the original quantized weights and activations by ignoring the higher-order bits, or quantize using the higher bit-width. As using the higher bit-width to quantize shadow weights and activations introduces noise to the model and might yield lower results, we try both approaches and only report a result if quantization using the higher bit-width gives better results.

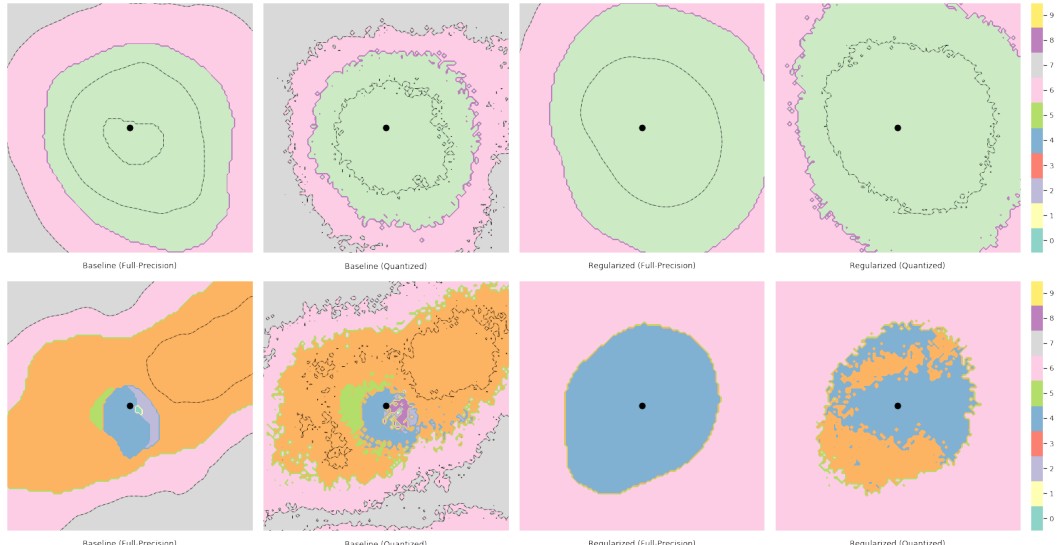

Figure 5: **Random cross sections of decision boundaries in the input space.** To generate these cross-sections, we draw a random example from the CIFAR-10 test set (represented by the black dot in the center) and pass a random two-dimensional hyper-plane $\subset \mathbb{R}^{1024}$ through it. We then evaluate the network's output for each point on the hyper-plane. Various colors indicate different classes. Softmax's maximum values determine the contours. The top row illustrates the difference between the baseline and the regularized VGG-like networks (and their quantized variants) when they all classify an example correctly. The bottom row depicts a case where the quantized baseline misclassifies an example while the regularized network predicts the correct class. We can see that our regularization pushes the decision boundaries outwards and enlarges the decision cells.

## 4.2 EFFECTS OF REGULARIZATION

In order to get a better understanding of our proposed regularizer, we first adopt the visualization method from Hoffman et al. (2019) and illustrate the effects that the quantization in general, and our method in particular, have on the trained classifier's decision boundaries. The result can be seen in Figure 5, where we empirically observe that the regularized networks "expands" its decision cells.

Secondly, we investigate in Figure 1 the $\ell_1$- and $\ell_2$-norms of the gradients for all CIFAR-10 test batches on the VGG-like model. We can observe that while the $\ell_2$-norms of the gradient are small in the unregularized model, the $\ell_1$-norms are orders of magnitude larger. Consequently, when fine-tuning the same model with our method, we see a strong decrease of the $\ell_1$-norm.

Finally, we investigate how the predictive distribution of the floating point model, $p(y|\mathbf{x})$, changes when we quantize either an unregularized baseline or a model regularized with our method, thus obtaining $q(y|\mathbf{x})$. We measure this discrepancy using the KL-divergence of the original predictive when using the predictive distribution of the quantized model, i.e. $D_{\mathrm{KL}}(p(y|x)||q(y|x))$, averaged over each test batch. Since our method improves robustness of the loss gradient against small perturbations, we would expect the per-class probabilities to be more robust to perturbations as well, and thus more stable under quantization noise. The result can be seen in Figure 2, where we indeed observe that the gap is smaller when quantizing our regularized model.

## 4.3 CIFAR-10 & IMAGENET RESULTS

The classification results from our CIFAR-10 experiments for the VGG-like and ResNet18 networks are presented in Table 1, whereas the result from our Imagenet experiments for the ResNet18 network can be found in Table 2. Both tables include all results relevant to the experiment, including results on our method, Defensive Quantization regularization, L2 gradient regularization and fine-tuning using the STE.

**Comparison to "Defensive Quantization"**  As explained in Section 3, Defensive Quantization (Lin et al., 2019) aims to regularize each layer's Lipschitz constant to be close to 1. Since the

| | VGG-like | | | | ResNet-18 | | | |
| --- | --- | --- | --- | --- | --- | --- | --- | --- |
| | FP | (8,4) | (6,4) | (4,4) | FP | (8,4) | (6,4) | (4,4) |
| No Regularization | 92.49 | 79.10 | 78.84 | 11.47 | 93.54 | 85.51 | 85.35 | 83.98 |
| DQ Regularization | 91.51 | 86.30 | 84.29 | 30.86 | 92.46 | 83.31 | 83.34 | 82.47 |
| L2 Regularization | 91.88 | 86.64 | 86.14 | 63.93 | 93.31 | 84.50 | 84.99 | 83.82 |
| L1 Regularization (Ours) | 92.63 | 89.74 | 89.78 | 85.99 | 93.36 | 88.70 | 88.45 | 87.62 |
| STE @ (8,4) | – | 91.28 | 89.99 | 32.83 | – | 89.10 | 87.79 | 86.21 |
| STE @ (6,4) | – | – | 90.25 | 39.56 | – | – | 90.77 | 88.17 |
| STE @ (4,4) | – | – | – | 89.79 | – | – | – | 89.98 |

Table 1: **Test accuracy (%) for the VGG-like and ResNet-18 models on CIFAR-10**. STE @ (X,X) indicates the weight-activation quantization configuration used with STE for fine-tuning. DQ denotes Defensive Quantization (Lin et al., 2019). For the No Regularization row of results we only report the mean of 5 runs. The full range of the runs is shown in Figure 4.

| | | Configuration | | |
| --- | --- | --- | --- | --- |
| | FP | (8,8) | (6,6) | (4,4) |
| No Regularization | 69.70 | 69.20 | 63.80 | 0.30 |
| DQ Regularization | 68.28 | 67.76 | 62.31 | 0.24 |
| L2 Regularization | 68.34 | 68.02 | 64.52 | 0.19 |
| L1 Regularization (Ours) | 70.07 | 69.92 | 66.39 | 0.22 |
| L1 Regularization (Ours) ($\lambda = 0.05$) | 64.02 | 63.76 | 61.19 | 55.32 |
| STE @ (8,8) | – | 70.06 | 60.18 | 0.13 |
| STE @ (6,6) | – | – | 69.63 | 11.34 |
| STE @ (4,4) | – | – | – | 57.50 |

Table 2: **Test accuracy for the ResNet-18 architecture on ImageNet**. STE @ (X,X) indicates the weight-activation quantization configuration used with STE for fine-tuning. In addition to the $\lambda$ we found through the grid-search which maintains FP accuracy, we also experimented with a stronger $\lambda = 0.05$ to show that (4,4) accuracy can be recovered at the price of overall lower performance.

regularization approach taken by the authors is similar to our method, and the authors suggest that their method can be applied as a regularization for quantization robustness, we compare their method to ours. As the experiments from the original paper differ methodologically from ours in that we quantize both weights and activations, all results on defensive quantization reported in this paper are produced by us. We were able to show improved quantization results using defensive quantization for CIFAR-10 on VGG-like, but not on any of the experiments on ResNet18. We attribute this behavior to too stringent regularization in their approach: the authors regularize *all* singular values of their (reshaped) convolutional weight tensors to be close to one, using a regularization term that is essentially a fourth power regularization of the singular values of the weight tensors (see Appendix C). This regularization likely inhibits optimization.

**Comparison to explicit $\ell_2$-norm gradient regularization** We consider the $\ell_2$ regularization of the gradient, as proposed by Gulrajani et al. (2017), as a generalization of the DQ regularization. Such regularization has two key benefits over DQ: 1) we can regularize the singular values without reshaping the convolutional kernels and 2) we impose a less stringent constraint as we avoid enforcing all singular values to be close to one. By observing the results at Table 1 and 2, we see that the $\ell_2$ regularization indeed improves upon DQ. Nevertheless, it provides worse results compared to our $\ell_1$ regularization, an effect we can explain by the analysis of Section 2.

**Comparison to quantization-aware fine-tuning** While in general we cannot expect our method to outperform models to which quantization-aware fine-tuning is applied on their target bit-widths, as in this case the model can adapt to that specific quantization noise, we do see that our model performs on par or better when comparing to bit-widths lower than the target bit-width. This is in line with our expectations: the quantization-aware fine-tuned models are only trained to be robust to a specific noise distribution. However, our method ensures first-order robustness regardless of bit-

width or quantization scheme, as explained in Section 2. The only exception is the 4 bit results on ImageNet. We hypothesize that this is caused by the fact that we tune the regularization strength $\lambda$ to the highest value that does not hurt full-precision results. While stronger regularization would harm full-precision performance, it would also most likely boost 4 bit results, due to imposing robustness to a larger magnitude, i.e. $\delta$, of quantization noise. Table 1 includes results for a higher value of $\delta$ that is in line with this analysis.

## 5 CONCLUSION

In this work, we analyzed the effects of the quantization noise on the loss function of neural networks. By modelling quantization as an $\ell_\infty$-bounded perturbation, we showed how we can control the first-order term of the Taylor expansion of the loss by a straightforward regularizer that encourages the $\ell_1$-norm of the gradients to be small. We empirically confirmed its effectiveness, demonstrating that standard post-training quantization to such regularized networks can maintain good performance under a variety of settings for the bit-width of the weights and activations. As a result, our method paves the way towards quantizing floating-point models "on the fly" according to bit-widths that are appropriate for the resources currently available.

## ACKNOWLEDGMENTS

We would like to thank Markus Nagel, Rana Ali Amjad, Matthias Reisser, and Jakub Tomczak for their helpful discussions and valuable feedback.

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

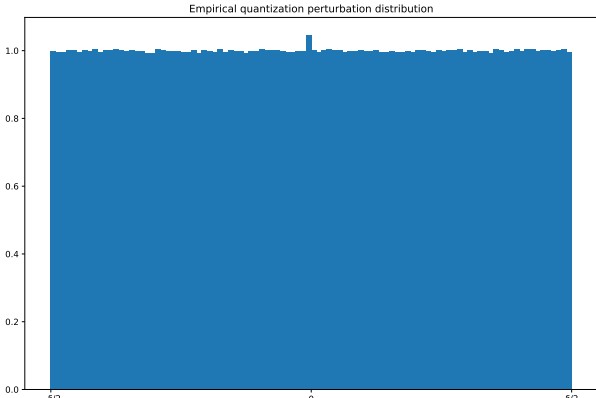

Figure 6: **Quantization noise is uniformly distributed.** In this plot we show the quantization noise on each individual weight in an ImageNet trained ResNet18 model. The noise is scaled by the width of the quantization bin for each weight quantizer. This plot shows that quantization noise is uniformly distributed between $-\delta/2$ and $\delta/2$.

## A    ROBUSTNESS ANALYSIS FOR QUANTIZATION PERTURBATIONS

In this section, we address two questions in more details, first regarding regularization of the $\ell_2$-norm of gradient and second regarding non-uniform quantization schemes.

We argued above that regularizing the $\ell_2$-norm of gradient cannot achieve the same level of robustness as regularization of the $\ell_1$-norm of gradient. We provide here another, more theoretical, argument. The following inequality shows how the $\ell_2$-norm of gradient controls the first-order perturbation:

$$\langle \mathbf{\Delta}, \nabla f(\boldsymbol{w}) \rangle \leq \|\mathbf{\Delta}\|_2 \|\nabla f(\boldsymbol{w})\|_2 \,.$$

This is a simple Cauchy-Shwartz inequality. Therefore, if the $\ell_2$-norm of the gradient is inversely proportional to the *power* of the perturbation, the first-order term is adequately controlled. However, using a theoretical argument, we show that the *power* of the $\ell_\infty$-bounded perturbation can blow up with the dimension as a vector $\mathbf{\Delta}$ in $\mathbb{R}^n$ with $\|\mathbf{\Delta}\|_\infty = \delta$ can reach an $\ell_2$-norm of approximately $\sqrt{n}\delta$. In other words, the length of the quantization noise behaves with high probability as $\Theta(\sqrt{n})$, which implies that the the $\ell_2$-norm of the gradient should be as small as $\Theta(1/\sqrt{n})$.

We show that this can indeed occur with high probability for any random quantization noise with the bounded support. Note that for symmetric uniform quantization schemes, quantization noise can be approximated well by a uniform distribution over $[-\delta/2, \delta/2]$ where $\delta$ is the width of the quantization bin. See Figures 6 for the empirical distribution of quantization noise on the weights of a trained network. Our argument, however, works for any distribution supported over $[-\delta/2, \delta/2]$, and, therefore, it includes asymmetric quantization schemes over a uniform quantization bin.

Consider a vector $\boldsymbol{x} = (x_1, \ldots, x_n)^T \in \mathbb{R}^n$ with entries $x_i$ randomly and independently drawn from a distribution supported on $[-\delta/2, \delta/2]$. We would like to show that $\|\boldsymbol{x}\|_2^2$ is well concentrated around its expected values. To do that we are going to write down the above norm as the sum of independent zero-mean random variables. See that:

$$\mathbb{E}\left( \|\boldsymbol{x}\|_2^2 \right) = \mathbb{E}\left( \sum_{i=1}^n x_i^2 \right) = n\mathbb{E}\left( x_1^2 \right) = \frac{n\delta^2}{12}\,.$$

Besides, note that $x_i^2 \in [0, \delta^2/4]$. Therefore $x_i^2 - \delta^2/12$ is a zero-mean random variable that lies in the interval $[-\delta^2/12, \delta^2/6]$. We can now use Hoeffding's inequality. To be self-contained, we include the theorem below.

**Theorem A.1** (Hoeffding's inequality, (Hoeffding, 1963))**.** *Let $X_1, \ldots, X_n$ be a sequence of independent zero-mean random variables such that $X_i$ is almost surely supported on $[a_i, b_i]$ for*

$i \in \{1, \dots, n\}$. *Then, for all $t > 0$, it holds that*

$$\mathbb{P}\left(\sum_{i=1}^{n} X_i \geq t\right) \leq \exp\left(-\frac{2t^2}{\sum_{i=1}^{n}(b_i - a_i)^2}\right) \tag{4}$$

$$\mathbb{P}\left(\left|\sum_{i=1}^{n} X_i\right| \geq t\right) \leq 2\exp\left(-\frac{t^2}{\sum_{i=1}^{n}(b_i - a_i)^2}\right) \tag{5}$$

Applying Theorem A.1 to our setting, we obtain:

$$\mathbb{P}\left(\left|\|\boldsymbol{x}\|_2^2 - n\delta^2/12\right| \geq t\right) \leq 2\exp\left(-\frac{2t^2}{n(\delta^2/4)^2}\right).$$

So with probability $1 - \epsilon$, we have:

$$\left|\|\boldsymbol{x}\|_2^2 - n\delta^2/12\right| \leq \left(\frac{n\delta^4}{32}\log(2/\epsilon)\right)^{1/2}.$$

Therefore, if the quantization noise $\boldsymbol{\Delta}$ has entries randomly drawn from a distribution over $[-\delta/2, \delta/2]$, then with probability $1 - \epsilon$, the squared $\ell_2$-norm of $\boldsymbol{\Delta}$, i.e., $\|\boldsymbol{\Delta}\|_2^2$, lies in the interval $\left[\frac{n\delta^2}{12} - \sqrt{\frac{n\delta^4}{32}\log(2/\epsilon)}, \frac{n\delta^2}{12} + \sqrt{\frac{n\delta^4}{32}\log(2/\epsilon)}\right]$. In other words, the length of the vector behaves with high probability as $\Theta(\sqrt{n})$. This result holds for any quantization noise with bounded support.

If the quantization bins are non-uniformly chosen, and if the weights can take arbitrarily large values, the quantization noise is no-longer bounded in general. As long as the quantization noise has a Gaussian tail, i.e., it is a subgaussian random variable, one can use Hoeffding's inequality for subgaussian random variables to show a similar concentration result as above. The *power* of the perturbation will, therefore, behave with $\Theta(\sqrt{n})$, and the $\ell_2$-norm of the gradient cannot effectively control the gradient. Note that nonuniform quantization schemes are not commonly used for hardware implementations, hence, our focus on uniform cases. Besides, the validity of this assumption about nonuniform quantization noise requires further investigation, which is relegated to our future works.

## B   SECOND-ORDER PERTURBATION ANALYSIS

We start by writing the approximation of $f(\cdot)$ up to the second-order term:

$$f(\boldsymbol{w} + \boldsymbol{\Delta}) = f(\boldsymbol{w}) + \langle \boldsymbol{\Delta}, \nabla f(\boldsymbol{w})\rangle + \frac{1}{2}\boldsymbol{\Delta}^T \nabla^2 f(\boldsymbol{w})\boldsymbol{\Delta} + R_3.$$

The worst-case second-order term under $\ell_\infty$-bounded perturbations is given by

$$\max_{\|n\|_\infty \leq \delta} \boldsymbol{\Delta}^T \nabla^2 f(\boldsymbol{w})\boldsymbol{\Delta}.$$

The above value is difficult to quantify for general case. We demonstrate this difficulty by considering some special cases.

Let's start with convex functions, for which the Hessian $\nabla^2 f(\boldsymbol{w})$ is positive semi-definite. In this case, the Hessian matrix admits a square root, and the second-order term can be written as:

$$\boldsymbol{\Delta}^T \nabla^2 f(\boldsymbol{w})\boldsymbol{\Delta} = \boldsymbol{\Delta}^T (\nabla^2 f(\boldsymbol{w}))^{1/2}(\nabla^2 f(\boldsymbol{w}))^{1/2}\boldsymbol{\Delta} = \left\|(\nabla^2 f(\boldsymbol{w}))^{1/2}\boldsymbol{\Delta}\right\|_2^2.$$

Therefore the worst-case analysis of the second-term amounts to

$$\max_{\|n\|_\infty \leq \delta} \boldsymbol{\Delta}^T \nabla^2 f(\boldsymbol{w})\boldsymbol{\Delta} = \max_{\|n\|_\infty \leq \delta} \left\|(\nabla^2 f(\boldsymbol{w}))^{1/2}\boldsymbol{\Delta}\right\|_2^2.$$

The last term is the mixed $\infty \to 2$-norm of $(\nabla^2 f(\boldsymbol{w}))^{1/2}$. As a reminder, the $p \to q$-matrix norm is defined as

$$\|\boldsymbol{A}\|_{p \to q} := \max_{\|x\|_p \leq 1} \|\boldsymbol{A}\|_q = \max_{\substack{\|\boldsymbol{x}\|_p \leq 1 \\ \|\boldsymbol{y}\|_{q^*} \leq 1}} \langle \boldsymbol{y}, \boldsymbol{A}\boldsymbol{x}\rangle =: \left\|\boldsymbol{A}^T\right\|_{q^* \to p^*}$$

where $p^*, q^*$ denote the dual norms of $p$ and $q$, i.e. satisfying $1/p + 1/p^* = 1/q + 1/q^* = 1$.

The worst case second-order perturbation is given by:

$$\max_{\|n\|_\infty \leq \delta} \boldsymbol{\Delta}^T \nabla^2 f(\boldsymbol{w}) \boldsymbol{\Delta} = \delta^2 \left\| (\nabla^2 f(\boldsymbol{w}))^{1/2} \right\|_{\infty \to 2}^2.$$

Unfortunately the $\infty \to 2$-norm is known to be NP-hard ((Hendrickx & Olshevsky, 2010); see Bhattiprolu et al. (2018) for a more recent study). As a matter of fact, if $f(\cdot)$ is positive semi-definite, and hence the function is convex, the problem above corresponds to maximization of convex functions, which is difficult as well.

For a general Hessian, the problem is still difficult to solve. First note that:

$$\max_{\|n\|_\infty \leq \delta} \boldsymbol{\Delta}^T \nabla^2 f(\boldsymbol{w}) \boldsymbol{\Delta} = \max_{\|n\|_\infty \leq \delta} \mathrm{Tr}\left( \nabla^2 f(\boldsymbol{w}) \boldsymbol{\Delta} \boldsymbol{\Delta}^T \right).$$

We can therefore replace $\boldsymbol{\Delta}\boldsymbol{\Delta}^T$ with a positive semi-defintite matrix of rank 1 denoted by $\boldsymbol{N}$. The worst case second-order perturbation can be obtained by solving the following problem:

$$\max_{\boldsymbol{N} \in \mathbb{R}^{n \times n}} \mathrm{Tr}\left( \nabla^2 f(\boldsymbol{w}) \boldsymbol{N} \right) \tag{6}$$
$$\text{subject to } \boldsymbol{N} \succeq 0$$
$$N_{ii} \leq \delta^2 \quad \text{for } i \in \{1, \ldots, n\}$$
$$\mathrm{rank}(\boldsymbol{N}) = 1.$$

The last constraint, namely the rank constraint, is a discrete constraint. The optimization problem above is therefore NP-hard to solve. To sum up, the worst case second-order perturbation cannot be efficiently computed, which poses difficulty for controlling the second-order robustness.

There are, however, approximations available in the literature. A common approximation, which is widely known for the Max-Cut and community detection problems, consists of dropping the rank-constraint from the above optimization problem to get the following semi-definite program:

$$\max_{\boldsymbol{N} \in \mathbb{R}^{n \times n}} \mathrm{Tr}\left( \nabla^2 f(\boldsymbol{w}) \boldsymbol{N} \right) \tag{7}$$
$$\text{subject to } \boldsymbol{N} \succeq 0$$
$$N_{ii} \leq \delta^2 \quad \text{for } i \in \{1, \ldots, n\}$$

Unfortunately this approximation, apart from being costly to solve for large $n$, does not provide a regularization parameter that can be included in the training of the model.

It is not clear how we can control the second-order term through a tractable term.

## C   DEFENSIVE QUANTIZATION IMPOSES A 4TH POWER CONSTRAINT ON SINGULAR VALUES

From basic linear algebra we have that

$$\|\boldsymbol{W}\|_2^2 = \mathrm{Tr}(\boldsymbol{W}^T \boldsymbol{W}) = \sum_i \sigma_i^2(\boldsymbol{W}),$$

i.e., the Frobenius norm is equal to sum of the squared singular values of $\boldsymbol{W}$. From this we can conclude that the regularization term $\|\boldsymbol{W}^T \boldsymbol{W} - \boldsymbol{I}\|_2^2$ introduced by Lin et al. (2019) thus equals

$$\|\boldsymbol{W}^T \boldsymbol{W} - \boldsymbol{I}\|_2^2 = \sum_i \sigma_i^2(\boldsymbol{W}^T \boldsymbol{W} - \boldsymbol{I}) = \sum_i \left| \sigma_i^2(\boldsymbol{W}) - 1 \right|^2,$$

and therefore imposes a 4th power regularization term on the singular values of $\boldsymbol{W}$. A softer regularization can be introduced by regularizing $\mathrm{Tr}(\boldsymbol{W}^T \boldsymbol{W} - \boldsymbol{I})$ instead.

# D  GRADIENT-PENALTY PROGRESSION IN NON-REGULARIZED NETWORKS

Optimizing our regularization penalty requires computing gradients of the gradients. While this is easily done by double-backpropagation in modern software frameworks it introduces overhead (as discussed in Section 4.1) and makes training slower. However, as the training progresses, the gradients in unregularized networks tend to become smaller as well, which is inline with our regularization objective. It is therefore not necessary to apply the regularization from the beginning of training. In Figure 7 we show examples of how the regularization objective naturally decreases during training. We also show how turning the regularization on in the final epochs where the regularization objective is oscillating can push the loss further down towards zero.

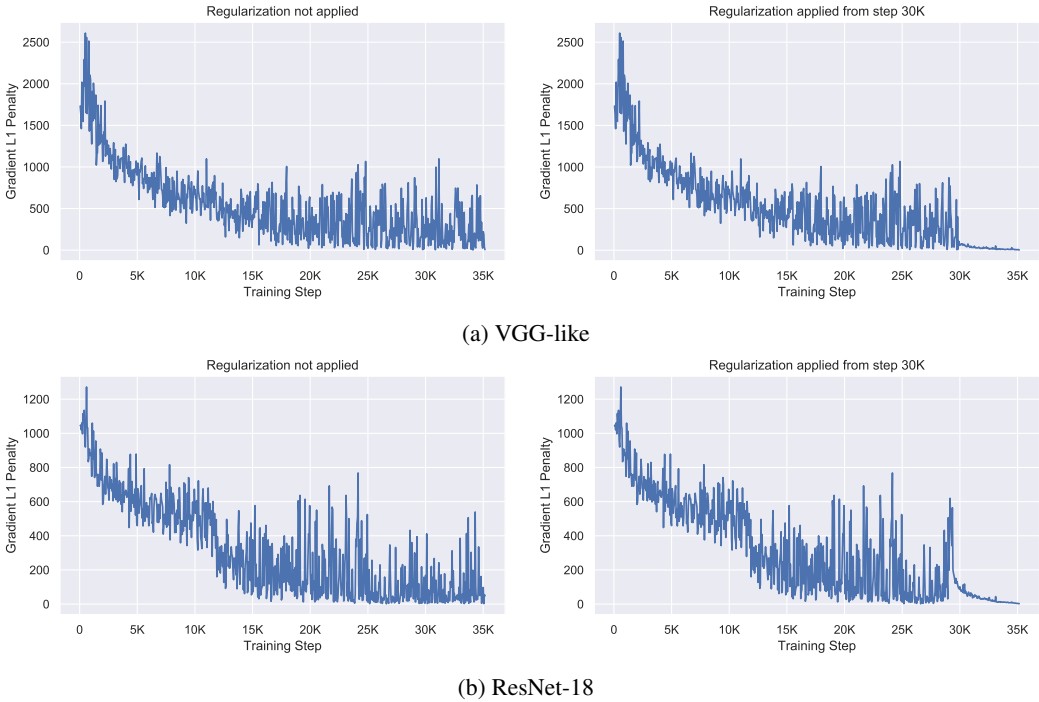

Figure 7: The gradients in unregularized networks tend to become smaller as training progresses. This means for large parts of the training there is no need to apply the regularization. The plots on the left show the regularization penalty in unregularized networks. The plots on the right show how turning on the regularization in the last 15 epochs of the training can push the regularization loss even further down.

# E  $\ell_\infty$-BOUNDED PERTURBATIONS INCLUDE OTHER BOUNDED-NORM PERTURBATIONS

Figure 8 show that the $\ell_\infty$-bounded perturbations include all other bounded-norm perturbations.

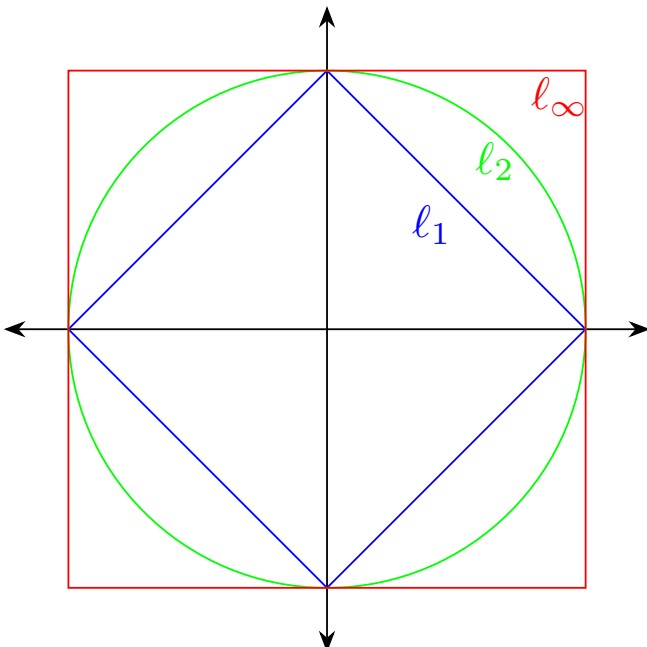

Figure 8: $\ell_\infty$-**bounded vectors include other bounded- norm vectors.** In this plot we show that the perturbations with bounded $\ell_p$-norm are a subset of $\ell_\infty$-bounded perturbations. For $p = 1, 2, \infty$, we plot the vectors with $\|\boldsymbol{x}\|_p = 1$.

