# OpenReview forum: "Gradient $\ell_1$ Regularization for Quantization Robustness"
_ICLR.cc/2020/Conference — Accept (Poster)_

### Official Review · AnonReviewer2 · 2019-10-16
**Official Blind Review #2**

**Rating:** 6

**Review:**

Summary: the authors propose a regularization scheme that is applied during regular training to ease the pose-training quantization of a network. Modeling the quantization noise as an additive perturbation bounded in \ell_\inf norm, they bound from above the first-order term of the perturbations applied to the network by the \ell_1 norm of the gradients. Their claims are also supported by experiments and qualitative illustrations.

Strengths of the paper:
- The paper is clearly written and easy to follow. In particular, section 2.1 clearly motivates the formulation of the regularization term from a theoretical point of view (reminiscent of the formulation of adversarial examples) and Figures 1 and 2 motivate the regularization term from a practical point of view. I found Figure 5 particularly enlightening (the regularization term "expands" the decision cells).
- The method is clearly positioned with respect to previous work (in particular using \ell_2 regularization of the gradients)
- Experiments demonstrate the effectiveness fo the method.

Weaknesses of the paper:
- The link between the proposed objective and the sparsity could be made clearer: does this objective enforce sparsity of the gradients, the weights, and how does this affect training?


Justification of rating:
The paper clearly presents a regularization method to improve post-training quantization. The approach is motivated both from a theoretical point of view and from a practical point of view. The latter aspect is of particular interest for the community. The claims are validated by a limited set of experiments that are seem nonetheless well executed.



**Experience Assessment:**

I have published one or two papers in this area.

**Review Assessment: Checking Correctness Of Derivations And Theory:**

I assessed the sensibility of the derivations and theory.

**Review Assessment: Checking Correctness Of Experiments:**

I carefully checked the experiments.

**Review Assessment: Thoroughness In Paper Reading:**

I read the paper at least twice and used my best judgement in assessing the paper.

---

> ### Author Response · Authors · 2019-11-12
> **Response to Review #2**
>
> We would like to thank the reviewer for the encouraging words and comments.
>
> While sparsity of the gradients is not something that we explicitly target, it is indeed a by-product of the objective. Intuitively our regularization imposes a constraint on the model that encourages it to be insensitive to bounded perturbations, in the first order sense. It should be noted that we recover weight sparsity when we adopt a linear model, as the L1 norm of the gradient is equivalent to the L1 norm of the weights.

---

### Official Review · AnonReviewer3 · 2019-10-23
**Official Blind Review #3**

**Rating:** 6

**Review:**

This paper models the quantization errors of weights and activations as additive l_inf bounded perturbations and uses first-order approximation of loss function to derive a gradient norm penalty regularization that encourage the network's robustness to any bit-width quantization. The authors claim that this method is better than previous quantization-aware methods because those methods are dedicated to one specific quantization configuration.

The derivation of the proposed method is not complex but I like the idea that models quantization error as additive perturbation in this context and how it eventually connects with gradient penalty that's widely used in GAN training and adversarial robustness.

Questions:

1. What is the capital N in the time complexity of gradient computation in Sec. 4.1? The authors should discuss in details the time complexity of the proposed regularization well because this is an essential problem of the regularization, which involves double back-propagation and should be computationally heavy. For the same reason, I'd like to see the training time comparison, and more results with deeper networks.

2. Compared to STE, one of the quantization-aware methods, the proposed method is not very competitive even in the setting when a STE network, which is specially trained for 6,6 bits but quantized to 4,4 bits, can outperforms the proposed method. This contradicts with the claimed strength of the proposed method. Will it be better when we regularize more, if we want the model to perform well when quantized to 4,4 bits? It would be better if there is a set of experiments of different regularization hyperparameters.

***********************

Update: I'd like to keep my score after reading the authors' response to all reviewers. I think the authors do address some questions but the paper still has some weakness in terms of performance.

**Experience Assessment:**

I have published one or two papers in this area.

**Review Assessment: Checking Correctness Of Derivations And Theory:**

I carefully checked the derivations and theory.

**Review Assessment: Checking Correctness Of Experiments:**

I carefully checked the experiments.

**Review Assessment: Thoroughness In Paper Reading:**

I read the paper thoroughly.

---

> ### Author Response · Authors · 2019-11-12
> **Response to Review #3**
>
> We would like to thank the reviewer for the constructive comments.
>
> Your question about capital N is fair. N, in this case, is (roughly) the number of elements w.r.t. which we are computing the gradient (e.g. weights in the case of regular backprop). The point we were trying to make is that, while this is a second-order method, we do not need to compute the full Hessian w.r.t. the weights, which would have O(N^2) time and space complexity in the number of weights.
>
> To be more exact: auto-differentiation [1] of a function  "f:  R^n --> R^m", where "f" contains "E" elementary operations, requires O(m x C x E) time, where "C" is a fixed constant. The gradient_L1 penalty is a function "p: R^N --> R", where N is the number of elements in the gradient. This function contains O(N) elementary operations to compute the L1 norm. The function to compute the loss gradient contains O(N) elementary operations as well, one for every node in the original forward computation graph. Thus, from the formula above, the complexity of computing the gradient w.r.t. the gradient L1 norm is O(2xCxN). Since 2xC is a constant that does not depend on the input, the complexity is O(N). We have updated the paper (Section 4.1) to make all of this clearer and provide more details on the complexity of the algorithm.
>
> We have also added the new "Appendix E" to provide justification for enabling the regularization only in the final stages of the training. The appendix depicts the progression of the regularization objective in unregularized networks. We show that the regularization loss becomes smaller (up to a point) during training with no regularization and therefore we can apply the regularization when the regularization loss has plateaued and is oscillating. We have also added wall-time timing measurements of the overhead in Section 4.1of the draft.
>
> With regards to performance at (4,4) bits and comparison to STE you are absolutely right that this is related to the strength of the regularization. As we discuss in our reply to Review #1 our main criteria for choosing lambda was maintaining the accuracy of the unquantized model. We have now run more experiments with larger values for lambda and that indeed results in improved performance in the (4,4) case, albeit at the cost of overall lower accuracy across all bit-width configurations. We have updated the paper to include this result (Table 2, Section 4.2).
>
> [1] Atilim Gunes Baydin, Barak A Pearlmutter, Alexey Andreyevich Radul, and Jeffrey Mark Siskind. "Automatic differentiation in machine learning: a survey.", Journal of machine learning research

---

> ### Author Response · Authors · 2019-11-15
> **Follow-up Response to Review #3**
>
> Given the reviewer's comment on deeper models we have started experimenting with MobileNet-v2 on ImageNet as well. While not necessarily a deeper architecture, MobileNet has a more demanding backward path and could be a good test for our proposed regularization. Our initial experiments do show promising results of 50.2% top-1 accuracy in the (8-bit weights, 4-bit activations) configuration as opposed to 0.07% for vanilla post-training quantization, and 59% for fine-tuning using STE. However, given that MobileNet architecture is very sensitive to choices such as per-channel quantization vs. per-layer quantization, and BatchNorm folding, we would like to run more extensive tests before including results in the final revision.

---

### Official Review · AnonReviewer1 · 2019-10-27
**Official Blind Review #1**

**Rating:** 6

**Review:**

This paper shows that if we add L1 regularization on gradient in the training phase, the obtained model can achieve better post-training quantization performance since it is more robust to Linf perturbation on the weights. I like the intuition of the paper but there are several weaknesses:

1. The main concern is that the proposed method cannot outperform quantization-aware fine-tuning. This probably limits the application of the method --- it will only be used when there's not enough time budget for quantization-aware fine tuning for each specific choice of #bits. It will be good if the authors can discuss in what practical scenario their algorithm can be applied.

2. The method is only tested under uniform symmetric quantization. I believe to demonstrate that the L1 regularized models are indeed easier to be quantized, we need to test it on several different kinds of quantizations.

3. I have concerns about the hyper-parameter selection for lambda. The authors mentioned that lambda is chosen by grid-search, but what's the grid search criteria? In other words, are the hyper-parameters trying to minimize the validation error of the "unquantized model", or they are minimizing the validation error of the "post-quantized model"?

4. Some minor suggestions:

- The current paper uses boldfaced n as perturbation which is quite confusing (since small n is the dimension). I would suggest to replace it by something else, e.g, \Delta.

- Section 2.3 seems redundant. It's clearly that L1 regularization is better given it's the dual norm of Linf, so clearly it's better than L2 norm. You have proved L2 is not good anyway in experiments.

===========

After seeing the rebuttal, my concerns about the parameters have been well addressed. Also, I agree with the authors that there are use cases for post quantization, and personally I think post quantization is much easier to do in practice than quantization-aware training. However, this is quite subjective so the fact that the proposed method doesn't outperform quantization-aware training is still a weakness of the paper.

I would like to slightly raise the score to borderline/weak-accept. I hope the authors can have some experiments on non-uniform quantization if the paper is being accepted; I really think that will demonstrate the strength of the method. People will likely to use this method if it can consistently improve many different kinds of post quantization.

**Experience Assessment:**

I have published one or two papers in this area.

**Review Assessment: Checking Correctness Of Derivations And Theory:**

I carefully checked the derivations and theory.

**Review Assessment: Checking Correctness Of Experiments:**

I carefully checked the experiments.

**Review Assessment: Thoroughness In Paper Reading:**

I read the paper thoroughly.

---

> ### Author Response · Authors · 2019-11-12
> **Response to Review #1**
>
> We would like to thank the reviewer for the careful review and useful comments.
>
> Regarding the comparison to quantization-aware training, we would like to emphasize that our proposed method is not meant to serve as a direct substitute for quantization aware fine-tuning. As the reviewer rightly pointed out quantization-aware training/fine-tuning can often achieve better results for the specific target bit-width. Having said that, we believe there are interesting practical applications where quantization-robust models are more appropriate. For example, we can consider the task of using a neural network as part of a mobile application. In such cases, one might be interested in automatically constraining the computational complexity of the network such that it conforms to specific battery consumption requirements, e.g. using a 4-bit variant when the battery is less than 20\% but the full precision model when the battery is over 80\%. In these cases, we can quantize to a specific bit-width on-the-fly without worrying about fine-tuning and without having to store multiple (potentially large) quantized models on device. Another challenge with post-training fine-tuning of models is access to the training data which can be challenging in some scenarios e.g. due to GDPR regularizations. We have uploaded an updated version of the paper to clarify such potential use-cases and applications.
>
> Regarding quantization schemes other than uniform symmetric quantization, it should be noted that our proposed method works equally well for asymmetric quantization schemes. Our theoretical derivations hold as long as the quantization noise has bounded support, even if it is not uniformly distributed. We have revised our text in Section 2.3 and the supplementary materials to reflect this issue. We have also moved some of the discussion in Section 2.3 to the supplementary materials as suggested by the reviewer. Non-uniform quantization schemes are currently less hardware-friendly and have limited applicability. Therefore, the focus of our research has been on uniform quantization schemes. We have updated Appendix A to discuss non-uniform cases. Our analysis holds for certain situations in which non-uniform quantization is used. However, a general answer to these questions will require more research and is left for future work.
>
> The reviewer's comment on the hyper-parameter selection is a fair point and should have been made clearer in the paper. Our criteria for choosing $\lambda$ was: the highest value of $\lambda$ within our search space that does not affect the accuracy of the \emph{unquantized} model, i.e. we did not want regularization to cause any degradation in the accuracy of the model in the normal mode, while maximally regularizing the model. We did not do any quantization for validation or hyper-parameter tuning. Furthermore, as discussed in Section 4.1 we only apply regularization in the final stages of the training (we have now updated the paper with the new Appendix E to include evidence and justification for this), however, we do track the regularization term during the training. This enabled us to have a rough estimate of the scale of regularization term with respect to the cross-entropy term. We then performed the grid search over a few points above and below the scale value that would bring regularization term to the same level as the cross-entropy. We have now updated Section 4.1 to make this clearer.
>
> Lastly, since the initial submission of the paper, we have been running more experiments with different values of lambda. One motivation for these experiments was to see if we can recover the (4,4) performance in ResNet-18 on ImageNet. We have updated the Table 2 to include this additional result for the same architecture but with larger lambda. It shows that larger values for lambda do indeed allow much better performance at (4,4) but at the cost of overall accuracy degradation across all quantization targets.
>
> Re. the notation: This is a good suggestion. Our updated draft uses adapts the suggestion notation.
>
> Re. the redundant section: We have incorporated your comment by moving most of Section 2.3 into an appendix.

---

### Decision · Program_Chairs · 2019-12-19

**Decision:**

Accept (Poster)

**Comment:**

Reviewers uniformly suggest acceptance. Please take their comments into account in the camera-ready. Congratulations!